# Exploring factors contributing to adolescent suicide using psychological autopsy - A scoping review

Josna Soyuz[1‡], Teddy Andrews Jaihind Jothikaran[1‡*], K. K. Sakkir[2],
Anish V. Cherian[3], Lena Ashok[1], Varalakshmi Chandrasekaran[2], Boby Augustin[4],
S. Elstin Anbu Raj[5‡]

1 Department of Social and Health Innovation, Prasanna School of Public Health, Manipal Academy of Higher Education, Manipal, Karnataka, India, 2 Department of Global Public Health Policy and Governance, Prasanna School of Public Health, Manipal Academy of Higher Education, Manipal, Karnataka, India, 3 Department of Psychiatric Social Work, National Institute of Mental Health and Neurosciences(NIMHANS), Bengaluru, Karnataka, India, 4 Department of Social Work, Navajyothi college, Cherupuzha, University of Kannur, Kannur, Kerala, India, 5 Department of Health technology and Informatics, Prasanna School of Public Health, Manipal Academy of Higher Education, Manipal, Karnataka, India

‡ These authors contributed equally to this work.
* teddy.andrews@manipal.edu

## Abstract

### Background

Adolescent suicidality and associated behaviours are an alarming global public health concern. Understanding its underlying causes and trajectories necessitates the urgent development of targeted prevention strategies. Psychological autopsy (PA) methods provide an in-depth understanding of the complex risk factors leading to suicide in this vulnerable population.

### Objective

This scoping review aims to explore and synthesise the major risk elements contributing to adolescent suicide and to propose a synthesis of the trajectories leading to adolescent suicide.

### Methods

A methodological search was carried out to find psychological autopsy studies centered on adolescent subjects. The selection criteria were fulfilled by 15 papers, which were scrutinized with the help of a thematic synthesis method and extracted the main themes to create an overview of the scope and character of the risk factors found.

**Data availability statement:** All relevant data are within the paper and its Supporting Information files.

**Funding:** The author(s) received no specific funding for this work.

**Competing interests:** The authors have declared that no competing interests exist.

## Results

Four major themes emerged: (1) Individual risk factors, including psychiatric disorders, substance use, and previous self-harm; (2) Familial factors, such as parental mental illness, abuse, neglect, and family dysfunction; (3) Life events, including academic failure, interpersonal conflicts, and childhood trauma and (4) Environmental risk factors, such as access to lethal means, media influence, and lack of mental health support. The findings highlight the complex interplay of psychological, social, and contextual variables in adolescent suicide.

## Conclusion

Psychological autopsy studies offer valuable insights into the risk factors for adolescent suicide. A multidimensional approach addressing individual vulnerabilities, family dynamics, and broader social environments is essential for effective prevention efforts. Further research is needed to strengthen the evidence base and guide policy and intervention development.

## Introduction

Suicide represents a profound global public health crisis, accounting for 1.4% of all deaths worldwide and an estimated 703,000 lives lost annually [1]. The devastating impact extends far beyond the individual, deeply affecting families, societies, and economies. Alarmingly, suicide is the second leading cause of death among individuals aged 15–29 years [2].Each completed suicide is estimated to prompt at least 25 additional suicide attempts and profoundly impacts approximately 135 individuals with intense grief and other emotional challenges. Studies indicate that over 20 million people exhibit suicidal behaviour annually [3].

While suicide can affect individuals across all life stages, it constitutes a significant percentage of deaths among adolescents globally [1]. The fatal and non-fatal manifestations of suicidality in adolescents are pressing public health concerns, with a reported lifetime prevalence of 29.9% for suicidal ideation and 9.7% for suicidal behavior, particularly within the 12–17 age group worldwide [4] Tragically, around 46,000 adolescents aged 10–19 die by suicide globally each year, equating to one child lost to suicide every 11 minutes [5].

Adolescence, a foundational period of life, is characterised by significant physical, psychological, social, and moral development, rendering this age group particularly complex and vulnerable [6]. During this phase adolescent develop self-awareness and identity, while facing increasing obligations and expectations of the self and others as they are trying to figure out their space and roles in different systems around them, such as family, society and peers. Such dynamic interactions usually bring a lot of instabilities and numerical changes in adolescents' lives. [7]. The inexperience and inability to effectively navigate these fluctuations often contribute to mental health problems that can have long-lasting consequences for these emerging individuals

[8]. According to the Global data of 2019, approximately 166 million adolescents, of all genders, are experiencing mental health conditions, predicting that one in seven adolescents is experiencing mental health problems [9].

Understanding the diverse aspects of suicide, particularly among young people, remains the core for the development of coping and prevention strategies. One primary method to achieve this is through the use of a psychological autopsy (PA). PA introduced by Shneidman (1981) is an agreeable method to explain the process of determining a fatality of suicide. It typically involves retracing the death-causing event, delineating the death situation—suicide intent—and collecting information as possible about other related risk factors [10], Initially designed to be a tool for police inquiries to understand the details of confused deaths [11], PAs have become an acknowledged, significant research tool for the study of the risk factors of completed suicides now [12]. The technique significantly simplifies the complex and convoluted nature of suicide [13] by in-depth and step-wise unravelling the life of the deceased to find such psychological, social, and environmental factors that may have triggered their death [12].

Data about PAs is usually collected by interviewing the family members, friends, medical / therapists or any significant individuals of the deceased who can provide in-depth, relevant information about the individual died by suicide. Along with a review of personal and medical records in parallel [14,15]. This The comprehensive method opens up possibilities to recognize and comprehend the circumstances that have led to death, such as suicidal motives, sociological and familial parameters, and self-destructive behaviour [16]. Psychological autopsy provides a clearer perspective of the victim's profile and acts as an upfront and common method to accurately indicate the cause of death in completed suicides. [16]. Given the disturbingly high rates of adolescent suicide and the density of its contributing factors, a scoping review utilizing insights gathered from psychological autopsy studies is indispensable to broadly understand the landscape of adolescent's suicide and identifying critical areas for future research and intervention. This scoping review aims to systematically explore the multifaceted risk factors contributing to the phenomenon of suicide among adolescents by addressing the question of how psychological autopsy studies characterize the contributing and precipitating factors of suicide among the adolescent population and, subsequently, to propose a conceptual framework for understanding the complexity and process of suicidality within this vulnerable population.

## Methods

The scoping review was conducted following Joanna Briggs Institute (JBI) scoping review guidelines. In addition to this, we have ensured the quality of the methodology of the review and the reporting of the findings, the scoping review followed the PRISMA-SCR guidelines. The protocol was registered before the conduct of the review (10.17605/OSF. IO/8DS4M).

### Identifying research questions

The aim of the scoping review was to explore the risk factors of suicide among adolescents reported in the psychological autopsy study globally. The Population, Concept, and Context (PCC) framework was used for developing research questions and eligibility criteria (Table 1).

### Eligibility criteria

The research question was developed using the PCC framework and the inclusion exclusion criteria for the study were framed after developing the research questions and the details of the inclusion and exclusion criteria are mentioned in the Table 1.

### Search Strategy

The search strategy aimed to locate both published and unpublished studies. A three-step search strategy was utilized in this review. First, an initial search of PubMed was conducted to identify relevant articles on the topic. The text words

**Table 1. Eligibility criteria.**

| PCC | | Inclusion | Exclusion |
|---|---|---|---|
| Population | Adolescents | Aged between 10 and 19 | -Studies solely on children<br>-Studies solely on adults<br>-Studies where the adolescent age group is disaggregated |
| Concept | Psychological Autopsy Suicide | -The studies that used the psychological autopsy method to understand suicide<br>- studies published in English<br>-The studies from 2000 to 2024<br>-The studies confirming a clear verdict of suicide. | -Multiple publications of the same study<br>-Case reports of case series that do not follow a systematic psychological autopsy method.<br>- Studies not focused on clear suicide.<br>- Studies on other forms of death. |
| Context | Global | | |

contained in the titles and abstracts of relevant articles, as well as the index terms used to describe the articles, were used to develop a comprehensive search strategy. The major keywords included psychology, autopsy, suicidal ideation, and adolescent. The identified index terms were combined appropriately using suitable Boolean operators for building the search strategy. Finally, the search was carried out by various terms and MeSH terms such as ("psychological autopsy"[Text Word] AND ("suicidal ideation"[MeSH Terms] OR "suicidal ideation"[All Fields] OR "suicide"[MeSH Terms] OR "suicide"[All Fields]) AND ("adolescent"[MeSH Terms] OR "adolescent"[Text Word])) AND ((english[Filter]) AND (2000:2024[pdat])).The search was initially executed and PubMed and then translated to other databases, including Web of Science, SCOPUS, and ProQuest Central. The ProQuest databases, including theses and dissertations, were considered as grey literature. Followingly, Google Scholar was also considered for running the search. Additional articles were identified using hand searching at the inclusion stage. The search was restricted to the English language and articles published from 2000 to 2024 only.

## Study procedure and the selection of the studies

Following the search, all identified citations were collated and uploaded into Rayyan, and duplicates were removed. Following a pilot test, titles and abstracts were screened by two independent reviewers for assessment against the inclusion criteria for the review. Potentially relevant sources were retrieved in full, and their citations were documented. The full text of selected citations was thoroughly assessed against the inclusion criteria by two independent reviewers. Reasons for the exclusion of sources of evidence at full text that do not meet the inclusion criteria were recorded and reported in the scoping review. Any disagreements arising between reviewers at each stage of the selection process were resolved through discussion or with an additional reviewer. The results of the search and the study inclusion process were reported in a PRISMA flow diagram (Fig 1).

## Data extraction

Data were extracted from papers included in the scoping review by one reviewer using a data extraction sheet developed by the reviewers and cross-checked by the other reviewer. The data extraction sheet was pilot tested before the process. The data extracted included specific details such as author, title of the paper, year of publication, study design, study participants and demographic details, method of suicide and risk factors of suicide. This process helped to gather information about the participants, concept, context, study methods, and key findings relevant to the review questions. In case of any discrepancies, it was resolved through discussion with the other reviewers.

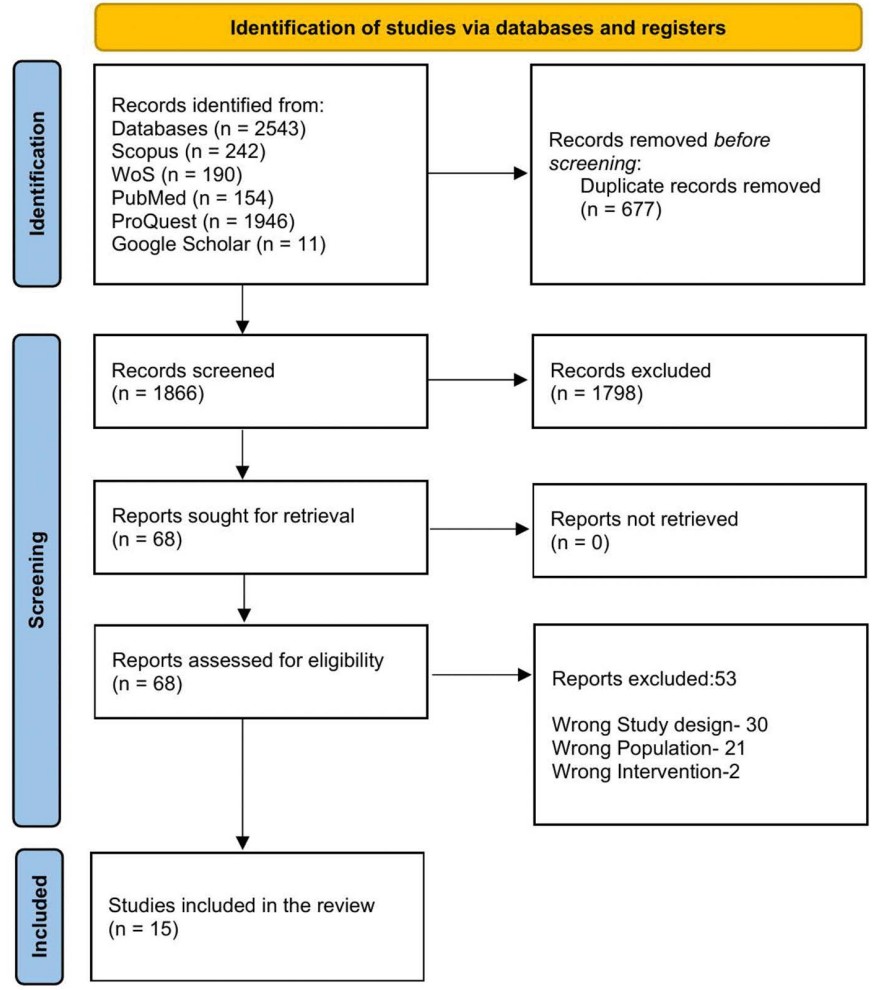

**Fig 1. PRISMA flow diagram.**

## Data analysis and presentation

The data extracted were presented using a narrative synthesis approach. The data were presented using tables and figures wherever appropriate. The retrieved data were analysed for codes and themes associated with adolescent suicide. This process involved firstly identifying codes reflecting the meaning of the description and similar codes were arranged together to generate major themes. Once the themes were identified, link and pattern between difference themes and risk factors for suicide among adolescents were generated. Since it was a scoping review we have not evaluated for the methodological rigor of the selected studies.

## Results

The search yielded a total of 2543 articles which were imported to Rayyan for further screening. After the deduplication process, a total of 1866 was considered for title and abstract screening. Out of which 68 were eligible for the full-text screening and finally 15 articles were deemed to fulfilling the inclusion criteria which were further considered for the review. The screening process has been detailed in Fig 1.

## Characteristics of the included studies

The scoping review encompasses 15 diverse range of psychological autopsy studies conducted among adolescent suicides from 2000 to 2024, spanning various countries such as Canada, Portugal, Sweden, Norway, the United Kingdom (2), the United States (2), Belgium (2), Israel, Korea, Mexico, the Netherlands, and Germany. Youths reviewed range roughly from age 10–21, with most studies focusing on mid-to late adolescence (10–19 years). These studies vary in methodology, encompassing case-control designs (e.g., matched suicide vs. community or violent-death controls), retrospective chart reviews, and multimethod qualitative interviews with informants such as family members, teachers, peers, and medical professionals. The sample size varies widely, from single-case analyses (18-year-old gifted student with ADHD) and small cohorts of 19–42 suicides (often paired with matched controls) to larger datasets of up to 140 adolescent suicides (with 131 controls) or 654 teacher-reported student suicides. Most of the studies utilised psychological autopsy techniques, incorporating structured or semi-structured interviews, often supplemented with forensic, medical, and school records.

The objectives of the studies include understanding psychiatric, psychosocial, familial, and contextual risk factors; distinguishing suicide from accidental death; and identifying patterns or warning signs preceding suicides. Common methodological characteristics include qualitative and mixed-method approaches, case-control comparisons, and in-depth informant interviews. Risk factors frequently explored across studies include psychiatric diagnoses (especially depression, ADHD, and adjustment disorders), impulsivity, previous suicide attempts, substance abuse, family history of suicide or mental illness, academic and relational stressors, and contextual and cultural pressures such as peer rejection or access to lethal means. The studies have emphasised the multifactorial nature of youth suicide and the importance of understanding both individual and environmental contributors through comprehensive postmodern investigations. The results were collated to develop a conceptual framework as depicted in Table 2.

## Factors affecting adolescent suicide

Based on 15 included global studies utilizing psychological autopsy conducted between 2000 and 2024, the factors contributing to adolescents suicide can be categorized into the following themes: 1) individual risk factors, 2) life events, 3) familial risk factors, and 4) environmental and contextual factors.

*Individual risk factors.* After examining the fifteen worldwide studies utilizing psychological autopsy, one of the most frequent and significant individual risk factors for suicide in adolescents was found to be the existence of mental disorders. Consequently, the various studies have been able to recognize major depressive disorder and other mood spectrum disorders as the main cause of suicidality in a significant number of cases. Fourteen out of sixteen studies, a significant number, indicated that there was a link between the association, thus, pointing to the far-reaching impact of this interaction [17 [18–26]. The repeated result here shows that the deepest root of a young person's susceptibility to suicide is the existence of serious mental health problems.

Besides mood disorders, four different studies also point out that ADHD can be an individual risk factor [20,24,27,28], signifying that Neurodevelopmental differences as well as difficulties in impulse control or emotional regulation, which could lead to an increased risk. Equally, adjustment disorder, characterized by significant emotional or behavioural symptoms in response to an identifiable stressor, was reported in two studies [22,27] suggesting that issues with coping in life changes or adverse events can also elevate suicide risk.

Eight studies identified that substance use, which had been encompassing alcohol and other unspecified substances, was one of the most significant concerns of adolescents suicidality [17–19,22,24,28–30]. The mentioned studies indicate that substance use, just for relieving the negative feelings or because the judgment is affected, is a key factor in the rise of suicidal thoughts and attempts among adolescents.

Various psychiatric and behavioral issues were among the factors that increased individual risk profiles. Increased screen time led to these new vulnerabilities; thus, an article dealing solely with the topic of Internet addiction [23], another

**Table 2.** Summary Table of Studies Included in the Scoping Review.

| Author Details | Sample details | Research Design | Focus area | Major Findings |
|---|---|---|---|---|
| Renaud et al., 2007 | *Country:* Canada *Age:* 11–18 *Sample size:* 55 (Male: 43, Female: 12). | *Method:* Cross-sectional Design: Case-control study *Collection:* Interviews | Individual risk factors. Family factors. | -Major Depression or any other diagnosis of mood spectrum, substance abuse, Diagnosis of ADHD, conduct disorders, physical illness, impulsive behaviours, previous attempts, tendency to aggression. -Family history of suicide and suicide attempts in first-degree relatives. |
| Mendes et al., 2015 | Country: Portugal Age: 10–18 Sample size: 17 cases | Method: Retrospective Collection: interview with family members. | Individual risk factor Cultural and contextual factors Life events | - Substance abuse, physical illness, impulsivity, psychiatric diagnosis, Physical illness, history of suicide attempts. -Easy access to lethal methods -breakup of a romantic relationship |
| Werbart Törn-blom et al., 2020 | Country: Sweden Age: 12–20 Sample size: 63 cases | Method: Case-control PA study Collection: Interview with family members and review of forensic data | Individual risk factors. Cultural and contextual factors. Life events. | -alcohol and substance abuse, borderline personality disorder, depressive disorders. - Odds of belonging, lower school results, and bullying at school. -Sexual assault, adverse childhood experiences |
| Freuchen et al., 2012 | Country: Norway Age: 15 years and younger Sample size: 84 cases | Design: Retrospective PA study with comparative design Collection: Interview with family members Review of records(police, forensic and hospital records) | Individual factors. Cultural and contextual factors. Life events Familial Factors | -Suicide threats, attempts, and self-harm behavior, mental health problems such as depression, anxiety, and ADHD diagnosis. Impulsive personality trait, absence from school before suicide. -Conflicting situations (parents, friends, police, and school), Bullying at school. -Loss of family members, disruption of romantic relationships -Parental divorce, remarried, or single parents |
| Fortune et al., 2007 | Country: UK Age: 14 years (mean age) Sample size: Not mentioned | Design: Retrospective PA study. Collection: Semi-structured interview with family members and document review. | Individual risk factors Cultural and contextual factors Life events Familial factors | Deliberate self-harm before death, evidence of an established psychiatric disorder, acute response to stressful life events, gambling, physical illness, impulsivity, highly sensitive to criticism. -Suicide in peers and bullying. -Breakdown in a close relationship, exposure to suicide behaviour, suicide in peers. - Family history of suicide, financial issues in the family, criminal history in the family, sibling rivalries, and attachment difficulties. |
| Goldstein et al., 2008 | Country: US Age: 13–19 Sample size:140 completed suicides and 131 controls. | Design: Case control PA study Collection: In-depth interview with family members | Individual risk factors | -Higher rate of sleep difficulties one week prior to suicide. |
| Cross et al., 2020 | Country: not mentioned. Age:18 Sample Size:1 | Design: Case study approach Collection: in-depth interviews with parents and teachers | factors Cultural and contextual factors Life events | Diagnosis of ADHD, diagnosis of mood and adjustment disorders, attempts of friends, social withdrawal, multiple suicide threats, and self-harm behavior. Lack of support from the system(school) and a less supportive environment, rejection from peers, multiple suicides in the community, availability of means to act, staying alone, and feeling of burdensomeness. |
| González-Castro et al., 2017 | Country: Mexico Age: 10–17 Sample Size: 28 cases | Design: Descriptive PA study Collection: Interview with first-degree relative. | Individual risk factors Contextual factors Familial Risk Factors | Break up with your girlfriend. - Alcohol consumption -Accessibility and availability of means -Parental alcoholism, dysfunctional families. |

*(Continued)*

**Table 2.** (Continued)

| Author Details | Sample details | Research Design | Focus area | Major Findings |
|---|---|---|---|---|
| Zalsman et al., 2016 | Country: Israel<br>Age:<br>Sample size: 70 high school students. | Design: Mixed-method PA study<br>Collection: Report files and interviews with significant people(parents, teachers, and relatives). | Individual risk factors<br>Contextual factors<br>Life events | -Substance abuse, lower school functioning, poor performance.<br>-Lower socioeconomic background, difficult psycho-social background, low capacity of the system (family, school) to identify red flags<br>-Stressful life events(didn't specify) |
| Lee et al., 2024 | Country: Korea<br>Age: 10–18 years<br>Sample size:654 cases of suicide. | Design: Descriptive retro-spective PA study<br>Collection: teachers' reports analysis | Individual risk factors<br>Contextual factors<br>Familial factors | - Mental disorder diagnosis (depressive disorders), internet addiction, and avoidant and submissive person-ality traits.<br>-Academic and peer relational stress.<br>-Parental divorce or separation, family history of suicide, parental mental illness, and parental discord during the developmental stage. |
| Mérelle et al., 2020 | Country: Netherlands<br>Age:10–19<br>Sample size:35 cases | Design: A multimethod PA study<br>Collection: interview with parents, siblings, teachers, and health professionals | Individual risk factors<br>Contextual factors<br>Life events<br>Familial factors | -Gender identity disorders, diagnosis of learning disabil-ities and ADHD, substance abuse, past attempts, social media addiction, and social withdrawal.<br>-Academic pressure and failures, financial difficulties, legal issues, lack of professional help, and bullying at school.<br>-Sexual abuse, running away from home, and break-down in romantic and peer relationships.<br>-Divorced parents, domestic violence, parental mental illness, disappointments in family. |
| Portzky et al., 2005 | Country: Belgium<br>Age: 15–19<br>Sample size: 19 suicide cases | Design: Retrospective PA study<br>Collection: Semi-structured interview with family members. | Individual risk factors<br>Contextual factors<br>Life events<br>Familial factors | - Behavioural and emotional problems, gender identity issues, alcohol and drug abuse, psychiatric diagnosis, the most common depressive disorder, and dependent personality disorder.<br>-Difficulties in making friends, Physical illness, and financial difficulties<br>-Academic failure, relationship breakdown, conflict with family members<br>-Unstable family environment, parental divorce, family history of suicide and psychiatric disorders, negative interactions and communications from parents. |
| Moskos et al., 2005 | Country: US<br>Age: 13–20<br>Sample size: 108 cases | Design: Retrospective psychological autopsy study<br>Collection: interviews with parents, siblings, friends, and other significant people. | factor<br>Contextual factors<br>Life events<br>Familial factors | Alcohol or other drug abuse, prior suicide attempts, struggles with sexual identity, psychiatric diagnosis, majorly depression, and ADHD.<br>Untreated mental illness<br>Suicide attempts in friends, childhood physical and sex-ual abuse, and breakdown in romantic relationships.<br>Divorced parents, family history of suicide<br>Mental illness, past suicide attempts. |
| Schmidt et al., 2002 | Country: Germany<br>Age: 10–20<br>Sample size: not specified | Study design: Retrospec-tive observational study<br>Collection: analysis of autopsy records | Individual risk factors<br>Contextual factors<br>Familial factors | Difficulties at school, problems within the friend circle. Relationship issues with a romantic partner and social isolation.<br>Disturbed family structure – Broken home, Mental illness in close relatives, suicidal behaviour within the family, conflict between family members, maltreatment. |
| Portzky et al., 2009 | Country: Belgium<br>Age: 15–19<br>Sample size: 19 cases | Study design: Matched case-control PA study<br>Collection: | Individual risk factors<br>Contextual factors<br>Life events<br>Familial factors | -Less communication, history of suicide attempts, his-tory of psychiatric treatment, personality disorder.<br>-Unhappiness at school, suicide exposure through the media.<br>-Relationship issues in the past year, academic failure, and criminal offenses.<br>-Parental divorce, conflict between parents and siblings, suicidal behaviour in family members. |

focusing on social network addiction [28], and a third one reporting developments in gaming, smartphones, and pornography-related disorders [25] have been identified. These results reveal how digital overuse or problematic digital consumption may influence teens' mental health. Moreover, several citations have mentioned that amongst the repeated observations of the temperamental peculiarities were the aggressive and impulsive natures of the individuals [17,18,20,21]. Such traits can turn into low decision-making abilities and raise the quantity of risky behaviors that one indulges in.

Further to this, peculiar attributes such as borderline personality traits, heightened sensitivity to criticism, and self-harm tendencies were revealed in four articles [19,21,27,31], pointing out substantial emotional dysregulation in such individuals. A report [17] also mentions the occurrence of conduct disorders, thus indicating the presence of severe behavioral problems. Minor personality traits, such as submissive, dependent, and avoidant personality traits [22,23], significantly pointing to reduced communication and social withdrawal, were identified in two studies [22,31]. It indicates interpersonal struggles and isolation in adolescents.

Several studies also pointed out different risk factors besides. The presence of suicidal history, which also included threats, was among the common findings in many studies [17,18,21,27,28,31], and this implies that it has a strong predictive value for future attempts. Besides, the research also revealed that physical ailments, especially HIV-related stigma, were some of the factors that led to a few cases [17,18,21], indicating that chronic health conditions could become a source of psychological suffering.

Behavioral indicators were identified alongside the main factors, for example, the absenteeism from school during the time before the attempt [20] and the acute reactions to stressful life events [21]. The participation in risky behaviors, such as gambling [21] and exposure to peer suicide [27], has considerably extended the theme of adolescents vulnerability. Additionally, a study singled out very severe sleep deprivation during the week before suicide [32], which could be considered a very close physical stressor.

Psychological autopsy investigations jointly reveal that the contributors to adolescents suicide are numerous and complex. It is observed that in most cases, as individual factors lead to suicide, the mental health coexisting with the mood disorders is always the main source of the hardest and most widespread individual risk factors.

*Life Events*- A substantial evidence of psychological autopsy studies consistently shows that the stressful life events play an evident role in significantly increasing the risk of adolescents suicide. These events can profoundly exacerbate existing vulnerabilities and precipitate suicidal ideation and behaviors.

One of the disturbing findings across three different studies is the devastating impact of adverse childhood experiences (ACEs), specifically childhood sexual abuse, on heightening the risk of developing suicidal thoughts [19,24,28]. This underlines the lifelong and severe psychological suffering that can influence adolescents suicidality.

In addition to early traumatic experiences, the immediate social environment of adolescents is often involved with multiple stressors. Troublesome peer and romantic relationships were frequently cited as substantial trigger for distress [20,21,24,27,28,31]. For adolescents, their social environment, including their interactions, is crucial for their development and self-esteem; struggles within this environment can lead to a profound sense of rejection, isolation, and hopelessness. Equally, academic problems, including failures that resulted in feelings of stigma and shame, were found to increase suicidal ideation profoundly [28]. The deepened burden of academic success, combined with the fear of failure and its social consequences, can create an overwhelming burden for some adolescents. One study explicitly found that exposure to suicide behaviors, especially the suicide of peers, can directly trigger suicidal ideation in vulnerable adolescents [21], emphasizing the risk of contagion.

Family-related life events were also identified as a significant contributing factor. The loss of family members [20], whether due to death or separation, can be traumatic for adolescents. Moreover, the challenges arising from parental divorce and remarriage [31], along with ongoing internal family conflicts, were recognized as significant stressors. In extreme situations, parental neglect and homelessness were also linked to the incidence of suicidal behavior [28], highlighting the dynamic part of a stable and supportive family atmosphere in shielding adolescent mental health.

*Familial factors-* As a primary system, the family plays a crucial role in shaping individual development. Factors such as parenting styles and practices, along with the quality of family relationships and atmosphere, and parental educational levels as well, are closely linked to a child's mental health course.

Adolescents raised within familial environments characterized by parental mental illness, domestic violence, or abuse face a significantly elevated risk of developing various mental disorders. Critically, a direct and concerning correlation exists between a family history of mental disorders and suicide-related behaviours in adolescents. Eight independent studies consistently identified a family history of suicide and, specifically, parental suicide, as robust and significant predictors of adolescent suicide [17,21,23–25,28,31]. This convincing mark highlights the collective influence of genetic disposition, learned behaviours, and the profound trauma connected with family suicide.

More than the direct impact of parental mental health issues, one study reported that sibling rivalries and attachment difficulties within the family, coupled with a familial criminal history, also contribute to an adolescent's suicide risk [19]. These scarcely cited components point out that strained intra-familial dynamics and exposure to criminal behaviour can influence family cohesion and support, increasing vulnerability.

Similarly, seven studies consistently reported that individuals originating from high-risk parental environments are particularly vulnerable to suicide. This broad category encompasses adolescents with divorced or remarried parents, children of single parents, those from overtly dysfunctional families, and individuals who have experienced significant trauma due to parental discord during their early developmental stages [20,22–24,28,31]. These findings collectively highlight how instability, lack of consistent parental presence, unresolved conflicts, and emotional distress within the family structure can deprive adolescents of the crucial emotional security and stable environment necessary for healthy psychological development, thereby increasing their vulnerability to suicidal behaviours.

*Environmental and contextual risk factors-*Beyond individual vulnerabilities, numerous studies consistently highlight the critical role of environmental and contextual risk factors in adolescents suicide, particularly those originating from the school environment. Bullying at school emerged as a prominent and dynamic risk factor, reported across several studies [19–21,28]. The sustained psychological distress and social isolation caused by bullying significantly contribute to an adolescent's vulnerability. Furthermore, academic pressures and hazards were frequently mentioned; declined academic performance, poor school results, and intense worries regarding academic failure were identified as factors that make adolescents susceptible to suicidality [19].

Adolescents also have to deal with a complicated social environment within their educational settings. Challenges such as insufficient support from the educational system, troublesome peer relationships, and strained romantic partnerships [31] contributing to existing vulnerabilities. Moreover, the struggle to be a part of peer circles, fit in with school groups, and the concerns of avoiding legal issues can further isolate vulnerable adolescents. The highly competitive nature of the school environment, coupled with heavy academic expectations, the gravity of parental pressure, and perceived interferences in educational quests, altogether contributes to the progress of potential suicidal ideation among adolescents [22,23,25,27,31]. These stressors highlight the need for supportive and healthy educational ecosystems.

Other than the school setting, various studies have identified additional sensitive contextual and environmental risk elements. The accessibility of lethal methods was repeatedly found to navigate the transition from suicidal ideation to an actual behavior, including attempts [18,27,29]. This emphasizes the need for immediate actions to be taken, such as means restriction strategies in suicide prevention efforts. Consequently, systemic failures within key support networks, such as schools, peer groups, and families, in recognizing and adequately responding to warning signs, further increase the risk [30]. This vulnerability is particularly noticeable when combined with limited access to professional mental health services [28] and untreated mental health issues [24], forming a dangerous gap in care.

The pervasive influence of media was also noted, with extensive media coverage of suicide drawn in as a significant contextual precipitant [31], suggesting the potential for contagion effects. Furthermore, adverse psychosocial circumstances, including socioeconomic hazards [28,30], social isolation, and a strong sense of perceived burdensomeness,

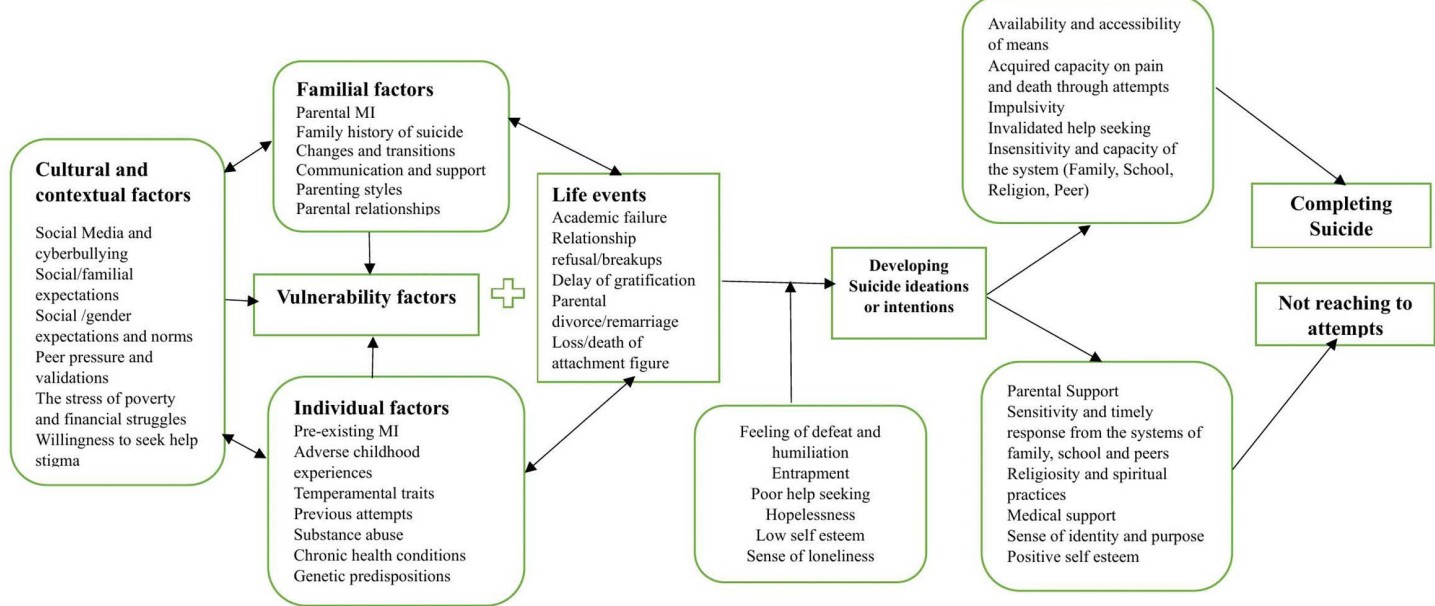

**Fig 2. Conceptual framework to understand the process of suicide among adolescents derived from these results.**

later increase an adolescent's vulnerability [19,24,30]. Adolescents living alone or lacking a fundamental sense of belong-ingness face elevated risk, as do those residing in communities that have tragically witnessed multiple suicide incidents [27]. These findings collectively underscore the reflective significance of both reducing access to lethal means and crit-ically strengthening detection, intervention, and support systems within familial, educational, and broader community settings to protect adolescents wellness (Fig 2).

The synthesis of the finding helped us to develop a conceptual model of the process of suicide among adolescents. Adolescent suicide risk follows a complex pathway, commencing from cultural and contextual antecedents, which affect underlying vulnerabilities such as mental illness in the family, past adverse experiences, personal temperament, and family relationships. This process then leads to critical life events, such as academic underachievement, relationship issues, or loss of significant attachment figures, leading to the development of suicide ideation, characterized by feelings of defeat, hopelessness, entrapment, and loneliness. At this critical stage, the process can lead to suicide completion, characterized by factors such as access to lethal means, increased pain tolerance from past suicide attempts, impulsive-ness, and ineffective help-seeking, or can be diverted from the cycle of suicide risk by protective factors such as robust parental support, timely and sensitive help from family, school, and peer networks, spiritual or religious activities, medical intervention, and the development of a positive concept of identity and self-worth.

## Discussion

This scoping review synthesized findings from 15 psychological autopsy studies conducted between 2000 and 2024 and offered a comprehensive understanding of factors contributing to adolescent suicide globally. One of the important find-ings across the reviewed studies is the significant role of individual risk factors, particularly psychiatric illnesses. Mental health conditions spanning from mood disorders to behaviour addiction like mobile and internet addictions were reported in the PA studies. Major depressive disorder and other mood disorders emerged as the most frequently reported individual risk factor, identified in fourteen studies. Attention-deficit/hyperactivity disorder (ADHD) was also noted as a probable risk factor in five studies. Substance use, including alcohol, was another commonly identified individual risk factor, appearing

in nine studies. This underscores the critical need for early identification and intervention for mental health conditions in adolescents.

Beyond formal diagnoses, temperamental characteristics such as aggression and impulsivity, borderline personality traits (including sensitivity to criticism and self-harm tendencies), and conduct disorders were identified as risk factors for suicide among adolescents. In addition to these, a history of prior suicide attempts, physical illness, school absenteeism, and gambling also increased the risk. These findings suggest that inherent personality traits and behavioural patterns, even in the absence of a full psychiatric diagnosis, can increase an adolescent's vulnerability to suicide.

Life events, especially Adverse Childhood Experiences (ACEs), play a critical role, acting as potential triggers or exacerbating existing vulnerabilities. Stressful life events, particularly childhood sexual abuse and those related to family, relationships, and school and studies, were frequently reported as a risk factor for suicide among the young population. Systematic review of life events reported a dose-response association between the number of ACEs and suicide related behaviours in young people, that is, more the number of ACEs higher the risk for suicide related behaviours [33, 34]. These findings could be used to develop suicide preventive strategies, for example, implementing interventions to reduce the risk of sexual abuse and other maltreatment, and early screening and identification of children with ACEs. Such strategies will be useful in reducing the occurrence of aversive experiences, and early identification could provide an opportunity to help victims deal with negative outcomes associated with these traumatic experiences.

Family history of suicide and mental illness has been consistently reported as a major risk factor for suicide and suicide related behaviours among adolescents. A case control study among adolescents revealed that negligent, affectionless-control bonding, insecure attachment, and stressful life events increased the risk for suicide, and parents' care and security were protective factors [35]. Children with a family history may be considered an at-risk group, and specific evidence-based prevention strategies need to be implemented for them. Adolescent-focused, parent-focused, and teacher-focused prevention strategies are recommended for the prevention [36]. Identifying youth at risk and providing treatment and support through a gatekeeper training program is effective in bringing a positive impact [36].

The review also highlighted the roles of contextual and environmental factors in increasing the risk of suicide among adolescents. Poor social support, inaccessibility of mental health services, school and study-related stress, and social isolation are some of them. Some of the recommendations for community-based prevention include strengthening economic supports, strengthening access and delivery of suicide care, creating protective environments, and promoting connectedness [37]. Our review showed that exclusive media coverage is implicated contextual precipitant. Sociological theories and research demonstrated that exposure to suicide and suicide relate news increases the risk of suicide [38,39]. Reporting suicide, especially of celebrities, made an impact on the general population [40,41]. Therefore, strict implementation of guidelines for media reporting and constant screening and monitoring of media reports will bring out hopeful results [42].

Most studies are published in European and North American countries, with limited representation of Asian studies, particularly Indian studies. Suicide, especially among adolescents, is an interplay of multiple precipitants, which include the cultural and sociopolitical background. Therefore, the review highlights the research gap and dearth of evidence of studies addressing suicide among adolescents from India.

## Implication for Policy, practice and future research

To address adolescent suicidality, a unified effort is necessary across various settings. Early detection through routine mental health screening, psychosocial assessment, and family awareness campaigns is essential. Likewise, school interventions such as mental health literacy, gatekeeper training for teachers, and peer support programs must be put into place. On the policy front, the focus should be on creating comprehensive suicide prevention strategies at national and regional levels. Barriers to mental health care need to be reduced, and stigma surrounding the issue must be diminished. Enhanced standardised surveillance is vital to guide interventions. Future research should explore long-term risk factors, evaluate the effectiveness of multidisciplinary prevention programs, and investigate the potential of new technologies in managing the issue.

## Limitations of this study

Psychological autopsy studies offer invaluable in-depth, retrospective insights into individual suicide cases; they inherently suffer from several methodological challenges. A primary concern is recall bias, as information is gathered from informants (family, friends, etc.) *after* the suicide has occurred, potentially leading to distorted or incomplete recollections influenced by grief, guilt, or a desire to protect the deceased's image. Furthermore, there's a lack of standardization across psychological autopsy methodologies globally, meaning the consistency and comparability of findings across the 15 included studies might be limited due to variations in interview protocols, data collection, and diagnostic criteria.

## Conclusion

This scoping review robustly underscores the complex and multifactorial nature of adolescent suicide risk, drawing comprehensively from 15 global psychological autopsy studies. It conclusively establishes psychiatric illness, particularly mood disorders, as the most consistently identified and critical individual risk factor, demanding primary attention in clinical assessment and intervention. Beyond individual vulnerabilities, the review highlights the profound influence of environmental and contextual stressors, notably bullying and academic pressures within school settings, and the pervasive impact of accessible lethal means.

Furthermore, it emphasizes the devastating role of stressful life events, including adverse childhood experiences and strained interpersonal relationships, in precipitating suicidal behaviors. Critically, the review illuminates the indelible influence of familial factors, with a family history of mental disorders and suicide emerging as significant predictors. Collectively, these findings advocate for a holistic approach to adolescent suicide prevention that transcends individual pathology to address systemic failures in support networks, improve access to mental health services, implement effective means restriction strategies, and foster stable and supportive familial and educational environments. While the review provides a comprehensive mapping of risk factors, future research incorporating quantitative synthesis and a focus on evidence-based interventions will further enhance the understanding and prevention of adolescent suicide.

## Supporting information

**S1 Table.** Preferred Reporting Items for Systematic reviews and Meta-Analyses extension for Scoping Reviews (PRISMA-ScR) Checklist.
(PDF)

## Author contributions

**Conceptualization:** Josna Soyuz, Teddy Andrews Jaihind Jothikaran, Anish V. Cherian, Lena Ashok, Varalakshmi Chandrasekaran, Boby Augustin, S. Elstin Anbu Raj.

**Data curation:** Josna Soyuz, K. K. Sakkir, S. Elstin Anbu Raj.

**Formal analysis:** Josna Soyuz.

**Methodology:** Josna Soyuz, Lena Ashok, Boby Augustin, S. Elstin Anbu Raj.

**Supervision:** Teddy Andrews Jaihind Jothikaran, Anish V. Cherian, Lena Ashok, Varalakshmi Chandrasekaran, Boby Augustin, S. Elstin Anbu Raj.

**Validation:** Teddy Andrews Jaihind Jothikaran, K. K. Sakkir, Varalakshmi Chandrasekaran, S. Elstin Anbu Raj.

**Writing – original draft:** Josna Soyuz.

**Writing – review & editing:** Teddy Andrews Jaihind Jothikaran, Anish V. Cherian, Varalakshmi Chandrasekaran, Boby Augustin.

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
