## [Decision Letter · Decision Letter 0]

5 Feb 2026

PONE-D-25-48263Exploring factors adding to adolescent suicide using psychological autopsy - a scoping reviewPLOS One

Dear Dr.  Soyuz,

Thank you for submitting your manuscript to PLOS ONE. After careful consideration, we feel that it has merit but does not fully meet PLOS ONE’s publication criteria as it currently stands. Therefore, we invite you to submit a revised version of the manuscript that addresses the points raised during the review process.

We look forward to receiving your revised manuscript.

Kind regards,

Massimiliano Esposito, M.D.

Academic Editor

PLOS One

Journal Requirements:

3. We note that your Data Availability Statement is currently as follows: “All relevant data are within the manuscript and its Supporting Information files”

Additional Editor Comments:

1. Title and Abstract

Original Title: "Exploring Factors Contributing to Adolescent Suicide Using Psychological Autopsy - A Scoping Review"

Suggested Revision: "Exploring Factors Contributing to Adolescent Suicide Using Psychological Autopsy: A Scoping Review"

Note: "Contributing to" is more formal than "adding to."

Abstract (Background): "...calls for an hour to develop..."

Suggested Revision: "...necessitates the urgent development of targeted prevention strategies."

Note: The expression "calls for an hour" is likely a literal translation from the Italian "è l'ora di," but it is not used in academic English.

Abstract (Objective): "...synthesis of how adolescents end up with suicide."

Suggested Revision: "...synthesis of the trajectories leading to adolescent suicide."

Note: "End up with suicide" is too colloquial.

2. Introduction

Original text: "Each completed suicide is believed to motivate at least 25 others to attempt suicide..."

Suggested revision: "Each completed suicide is estimated to prompt at least 25 additional suicide attempts..."

Note: "Prompt" or "trigger" are more appropriate technical terms than "motivate" in this context.

Identity Development: "This is the phase where adolescents develop their sense of self-awareness..."

Suggested revision: "During this phase, adolescents develop self-awareness and identity, while facing increasing obligations..."

Note: Rearranging the sentence improves fluency (avoid "the adolescents").

3. Methodology

Inclusion Criteria: "The studies analyzed with a clear verdict of suicide."

Suggested revision: "Studies confirming a clear verdict of suicide."

Search Strategy: "The combination of the final search terms was..."

Suggested Revision: "The final search string comprised..."

Study Selection: "Any disagreements that arise..."

Suggested Revision: "Any disagreements arising between reviewers..." (Use the participle for easier reading.)

4. Results

Individual Factors: "...one of the most recurring and most essential individual risk factors..."

Suggested Revision: "...one of the most frequent and significant individual risk factors..."

Note: In scientific contexts, "significant" or "prominent" are preferable to "essential."

Substance Use: "...referred to not only as alcohol but also other unmentioned substances..."

Suggested revision: "...encompassing alcohol and other unspecified substances..."

Technology: "...another focusing on the addictiveness of social networks..."

Suggested revision: "...another focusing on social network addiction..."

General Style Tips (Academic Tone):

Avoid contractions: Make sure you don't or can't (always use do not, cannot).

Use strong verbs: Replace compound verbs (phrasal verbs) with more formal single verbs (e.g., use investigate instead of look into, eliminate instead of get rid of).

Consistency of Terminology: You use both "completed suicide" and "suicide completion" in the text. Choose one and keep it consistent throughout the paper.

Articles: In several places, articles are missing or redundant (e.g., "the adolescents" should often be just "adolescents" when referring to the category in general).

Reviewer's Responses to Questions

**Comments to the Author**

1. Is the manuscript technically sound, and do the data support the conclusions?

Reviewer #1: Yes

2. Has the statistical analysis been performed appropriately and rigorously?

Reviewer #1: Yes

3. Have the authors made all data underlying the findings in their manuscript fully available?

Reviewer #1: Yes

4. Is the manuscript presented in an intelligible fashion and written in standard English?

Reviewer #1: Yes

5. Review Comments to the Author

Reviewer #1: Dear Editor, thank you for your invitation. Through a scoping review utilizing psychological autopsy studies, the research provides a concise and pertinent summary of teenage suicide. It successfully illustrates the multifaceted nature of suicide risk, and the topic is both pertinent and important. The methodological approach is appropriate, but the review's transparency and replicability would be strengthened with greater clarity regarding the search strategy, inclusion criteria, and geographic scope. The article could do a better job of explaining how thematic synthesis was carried out and whether the calibre of the included studies was evaluated, even though the four emergent themes are clearly categorized. Furthermore, there is a disconnect between the objectives and the results that are presented because, although the suggested conceptual framework is mentioned as an objective, it is not explained in the article. Although the conclusion is sound, more precise implications for practice, policy, and future research would be beneficial. It is informative and cohesive overall, but more methodological detail would improve it.

With regards,

Reviewer

6. PLOS authors have the option to publish the peer review history of their article (what does this mean?). If published, this will include your full peer review and any attached files.

Reviewer #1: **Yes:**Gyanesh Kumar Tiwari

---

## [Author Response · Author response to Decision Letter 1]

28 Mar 2026

A separate file has been uploaded responding to the specific reviewer and editor comments.

---

## [Editor Report · Decision Letter 1]

9 Apr 2026

Exploring factors contributing to adolescent suicide using psychological autopsy - a scoping review

PONE-D-25-48263R1

Dear Dr. Soyuz,

We’re pleased to inform you that your manuscript has been judged scientifically suitable for publication and will be formally accepted for publication once it meets all outstanding technical requirements.

Kind regards,

Massimiliano Esposito, M.D.

Academic Editor

PLOS One

---

## [Editor Report · Acceptance letter]

PONE-D-25-48263R1

PLOS One

Dear Dr. Soyuz,

I'm pleased to inform you that your manuscript has been deemed suitable for publication in PLOS One. Congratulations! Your manuscript is now being handed over to our production team.

Kind regards,

on behalf of

Prof. Massimiliano Esposito

Academic Editor

PLOS One